# Protocol for the systematic review of the reporting of transoral robotic surgery

Barry G Main,[1,2,3] Natalie S Blencowe,[1,2] Noah Howes,[2,4] Sian Cousins,[1,2] Kerry N L Avery,[1,2] Alexander Gormley,[5] Phil Radford,[5] Daisy Elliott,[1,2] Benjamin Byrne,[1,2] Nicholas Wilson,[1,2] Robert Hinchliffe,[1,2,4] Jane M Blazeby[1,2,3]

[1]National Institute for Health Research Bristol Biomedical Research Centre, Bristol, UK
[2]Bristol Centre for Surgical Research, Population Health Sciences, Bristol Medical School, Bristol, UK
[3]University Hospitals Bristol NHS Foundation Trust, Bristol, UK
[4]North Bristol NHS Trust, Bristol, UK
[5]Bristol Dental School, University of Bristol, Bristol, UK

**Correspondence to**
Dr Barry G Main;
b.g.main@bristol.ac.uk

## ABSTRACT

**Introduction** Transoral robotic surgery (TORS) has been adopted in some parts of the world as an innovative approach to the resection of oropharyngeal tumours. The development, details and outcomes of early-to-later phase evaluation of this technique and the quality of evidence to support its adoption into practice have hitherto not been summarised. The aim of this review is to identify and summarise the early and later phase studies of, and evidence for, TORS and to understand how early phase studies report intervention development, governance procedures and selection and reporting of outcomes to optimise methods for using the Idea, Development, Exploration, Assessment, Long-term follow-up (IDEAL) framework for surgical innovation that informs evidence-based practice. The protocol has been written in line with the Preferred Reporting Items for Systematic Review and Meta-Analysis Protocols checklist.

**Methods and analysis** Electronic searches in OVID SP versions of Medline and EMBASE, the Cochrane Central Register of Controlled Trials and the Cochrane Database of Systematic Reviews from the start of indexing to 30 April 2017 will identify studies reporting TORS. At least two independent researchers will identify studies for inclusion. Two researchers will extract data from each paper. Studies will be categorised into IDEAL stages of study design from 'pre-IDEAL' to randomised controlled trials (stage 3). Data will be collected about the (1) novel intervention and criteria for modification, (2) governance arrangements and patient information provision, (3) outcome domains selected and reported and (4) quality of study design, conduct and reporting. Descriptive statistics and a narrative synthesis will be presented.

**Ethics and dissemination** The results of this systematic review will be presented at relevant conferences. The methods will be used to inform future reviews exploring other novel surgical innovations. The findings will be published in a peer-reviewed journal. This study does not require ethical approval.

## Strengths and limitations of this study

► This will be a comprehensive review of transoral robotic surgery and will track its innovative evolution since first published description to present day.
► Inclusion of all study types will allow identification of good and poor examples of the descriptions of innovative invasive procedures.
► The methods described are applicable to reviews of any innovative surgical or other invasive procedure.
► Lack of a sensitive literature search strategy may result in large numbers of abstracts to screen at the initial stages.
► Exclusion of papers not published in English may mean that important additional findings are missed.

## INTRODUCTION

There is a need to improve outcomes for patients with oropharyngeal cancer. While surgery offers a chance of cure, it carries major risks, and there can be long-lasting detrimental effects on function and quality of life.[1] Recent developments in treatment strategies have, therefore, been aimed at organ preservation as well as overall survival.[2] Such developments have included attenuation in the field and dose of radiotherapy delivery and the introduction of innovative, less destructive surgical approaches including transoral laser and robotic techniques.

Transoral robotic surgery (TORS) is a technique first described in 2005 for the treatment of a benign pharyngeal cyst.[3] It has since been applied to the resection of oropharyngeal cancers and the treatment of obstructive sleep apnoea, was approved by the Food and Drug Administration (FDA) for these indications in 2009 and was subsequently adopted into routine practice in several centres, particularly in the USA.[2] The National Health Service in England do not routinely commission TORS for the treatment of oropharyngeal or laryngeal cancer.[4] The technique involves robotically assisted excision of a tumour or other tissue by an experienced surgeon who operates a control console in the same room but remote from the patient. Putative benefits include less invasive surgical access, better in situ three-dimensional visualisation of the tumour, scaling of movement and elimination of hand tremor.[5] Reported disadvantages include lack of tactile feedback and high cost.[6 7] There appears, however, to be a lack of synthesised data about the effectiveness of

TORS. Factors associated with the development, evaluation and implementation of this innovation have not yet been collated, described or summarised. The extent to which key issues around understanding surgical innovations, including clinical indications for the innovation, which patients are offered the innovation, what information is given to patients to ensure appropriate informed consent in the context of an innovative treatment (and within what governance structure), how the intervention is performed, which modifications are required during development (and what criteria inform decision to modify an intervention) and which outcomes and adverse events are reported in order to document known and potentially unknown consequences of new procedures have been considered also require clarification.

The Idea, Development, Exploration, Assessment, Long-term follow-up (IDEAL) framework was developed in 2009 and proposed a prospective, stepwise means of evaluating innovative procedures.[8 9] It aims to help researchers and authors design and report studies of surgical innovation in a transparent way. Widespread adoption of the IDEAL framework by surgical innovators, researchers and journals has yet to be realised, and it is not known if the evolution of technologies such as TORS has been reported in accordance with this framework. An in-depth analysis of published literature on TORS will provide a case study of the reporting of a recently developed surgical intervention which is now offered as part of routine practice.

## OVERALL OBJECTIVE

This study aims to systematically identify and comprehensively summarise and critique studies reporting the introduction, evolution and evaluation of TORS for obstructive sleep apnoea and oropharyngeal cancer.

### Specific research questions

The novel intervention and criteria for modification

1. How is the novel intervention reported and criteria for modification described?
2. How were patients selected to receive the intervention? Did these criteria change over time?
3. Were consecutive patients considered for eligibility to be offered the novel intervention? If not, were reasons provided?
4. Were details of patients treated during the same time frame, and who did not receive the novel intervention, reported (including reasons why they were not offered the new treatment, or whether they declined it and what treatment they did receive)?
5. How was surgeon expertise/learning curve reported?
6. Do the studies map onto the IDEAL stages of evaluation of surgical innovation, and is there evidence that the IDEAL stage evolves over time?
7. How are the novel intervention (TORS) and cointerventions and/or comparators defined in the included studies?

8. What is the overall quality of reporting of the included studies as measured using validated tools for randomised and non-randomised studies?

Governance arrangements

9. How are ethical/governance considerations reported? For example, what descriptions of the process of obtaining the informed consent of patient participants are provided, if any? (note: the study protocols may be consulted in addition to the main report)
10. Were patients informed about the innovative nature of the procedure?

Outcome domains reported

11. Which clinical and patient-reported outcomes, and adverse events, are measured and reported in these studies? How are these outcomes defined and the time points for measurement selected? Did the studies report 'failures' as well as 'successes'?
12. Were outcomes relating to the operator and/or team reported?
13. Which economic (eg, cost effectiveness) or other outcomes relating to healthcare resource use (eg, operating time) were reported?

## METHODS
### Eligibility criteria

Studies will be included in the review if they meet the following inclusion criteria:

#### Participants

All patients (children and adults) being treated for benign or malignant oropharyngeal disease. In addition to clinical studies, preclinical animal and cadaveric studies will be included for review so that the translation from these to first-in-human studies can be mapped.

#### Intervention

'Transoral robotic surgery' or 'robotically assisted surgery' or 'robotic surgery for oropharyngeal cancer' or 'robotic surgery for obstructive sleep apnoea' or 'robotic surgery for benign oropharyngeal disease'. Studies will not be limited to a single type, or manufacturer, of robot, or a particular technique.

#### Comparator(s)

Alternative surgical approaches (eg, transoral laser surgery) and/or non-surgical treatments (eg, radiotherapy and/or chemotherapy). It is anticipated that in certain types of early phase studies, there will be no comparator group.

#### Outcome(s)

Clinical outcomes, patient-reported outcomes, adverse events, complications and early technical or process outcomes, reported criteria for modifying or withdrawing the intervention, resources use, cost and economic outcomes and reports of surgeons' experiences of the new procedure (where documented).

## Search strategy and study selection

Searches will be undertaken in OVID SP versions of Medline, and EMBASE, PubMed, SCOPUS, Web of Science, the Cochrane Central Register of Controlled Trials and the Cochrane Database of Systematic Reviews. The search strategy will be developed with the assistance of an information specialist. Search terms for TORS, surgery for oropharyngeal cancer, non-surgical treatments of oropharyngeal cancer, surgery for obstructive sleep apnoea and benign oropharyngeal disease (including cysts and papillomas) will be combined using the Boolean 'OR' operator (online supplementary appendix 1). Preclinical (in vitro, animal and cadaveric) studies will be included only where they have been cited in a clinical study. The search will span from the start of indexing to 1 July 2017. Reference lists of included studies will be searched for additional references. Citations will be collated using Endnote reference management software. Exclusion criteria will be (1) conference reports and abstracts and (2) studies of brain, skull base, lung, oral, hypopharyngeal, parapharyngeal, oesophageal, thyroid, parathyroid, laryngeal, nasopharyngeal, eye, salivary gland or skin cancers.

## Identification and selection of papers

A customised inclusion/exclusion form will be used to screen abstracts and to provide an audit trail. Titles and abstracts will be screened independently by three authors (BGM, AG and PR). Any conflicts not resolved by discussion between these authors will be referred to the study team for discussion. The full-text versions of papers retained after title and abstract screening will be downloaded for further assessment of their eligibility for inclusion. The review will be conducted in line with the Preferred Reporting Items for Systematic Reviews and Meta-Analyses (PRISMA) checklist.[10] A PRISMA flow diagram will be produced.

## Data extraction and management

Data will be extracted independently by at least two assessors for each paper (BGM, JMB, NW, AG and PR). A customised data extraction form will be used to collect relevant data from each paper (online supplementary file). Data of interest will include:

The novel intervention and criteria for modification

1. author, date of publication, country of origin of study, journal of publication and rationale for the study;
2. details of any funding, including from robot manufacturers and/or other declared or potential conflicts of interest;
3. participant demographics (age, sex and tumour details);
4. details of the type of centre undertaking the intervention, including any workload data reported. The number and grade of surgeon(s) operating, including their experience with TORS (eg, training received and evidence of preceptorship);
5. interventions, cointerventions, comparators and whether each was defined. Any definitions provided in the papers will be recorded verbatim;
6. in-depth details of any descriptions of the component parts of the intervention and any comparator (eg, 'incision', 'access' and 'dissection')[11] to explore if any specific components are subject to ongoing innovation;
7. study classification using a modified IDEAL classification tool. The original IDEAL framework does not come with guidance on how to apply it to the assessment of studies already conducted and published.[8] An adaptation of the IDEAL framework has been designed to permit its application to the retrospective analysis of published studies. It has been tested on a limited set of case studies, but is as yet unpublished.[12] This systematic review will use the new tool to analyse included studies and provide further pilot testing of its validity and reliability;
8. details of how the authors discuss and make conclusions related to their findings. For example, any descriptions of the need for further evaluation, the readiness of the intervention for adoption into routine practice or reasons for stopping the use of the intervention;
9. the overall quality of reporting will be assessed using an appropriate validated critical appraisal tool (eg, Consolidated Standards of Reporting Trials for randomised controlled trials (RCTs)).[13]

   Governance arrangements
10. information about governance approvals (ethics committee, Institutional Review Board (IRB), clinical effectiveness committee approval, conformite Europeene (CE) mark approval, FDA approval and National Institute for Health and Care Excellence approval) and whether these approvals were for the device in general or its use for the treatment of specific clinical indications;
11. information about how informed consent was obtained from patients, including whether or not they were informed about the innovative nature of the procedure (and if so, how? Eg, verbal or written information);
12. data about how many patients declined the intervention.

    Outcome domains reported
13. names of individual outcomes reported and whether these were specified as primary or secondary outcomes;
14. definitions provided (if any) for each of the outcomes;
15. names of any patient-reported outcomes measures (PROMs);
16. the patient-reported outcome (PROs) will be classified and the scales used within each will be recorded. Any ad hoc, study-specific, non-validated PROMs will be documented;

17. technical outcomes including issues such as intraoperative events and process outcomes relevant to the evaluation of innovative procedures;
18. cost and other economic outcomes.

Both randomised and non-randomised studies will be assessed for risk of bias using validated tools. RCTs will be assessed using the Cochrane Risk of Bias tool[14] to assess the (1) adequacy of sequence generation, (2) allocation concealment, (3) blinding of outcome assessors, (4) completeness of outcome data and (5) other sources of bias including selection and funding bias. Non-randomised studies will be assessed using the Risk of Bias in Non-Randomised Studies of Interventions tool[15] for bias due to: (1) confounding, (2) selection of participants into the study, (3) classification of interventions, (4) deviation from intended interventions, (5) missing data, (6) measurement of outcomes and (7) selection of the reported result.

### Data syntheses and statistical analyses

Data will be entered into a custom database. Any discrepancies in data extraction will be resolved by discussion with the entire study group.

The findings will be tabulated and, where appropriate, descriptive statistics performed. A narrative synthesis will summarise the findings of the review by organising data into descriptive themes. The analysis will be longitudinal (eg, does the IDEAL stage evolve over the time span of the review) and cross-sectional (eg, how do studies at a given time point differ?). This review will not aim to make conclusions about the relative effectiveness of TORS over other treatments for oropharyngeal cancer because the focus is on how innovative surgical procedures are reported in the scientific literature. Therefore, no meta-analyses will be performed.

### DISSEMINATION

The systematic review will be published in a peer-reviewed journal and presented at appropriate conferences (eg, methodological and surgical). This protocol will be adapted for the analysis of other innovative surgical and invasive procedures.

### CONCLUSION

This systematic review will provide important information about the quality of studies of robotic surgery for oropharyngeal cancer and benign oropharyngeal disease. It will summarise the reporting of the novel intervention from early phase to any later phase studies published and available for review. It is intended to apply the methods described in this protocol to the analyses of reporting of other novel surgical interventions. The findings will inform further work that will aim to improve studies of innovative surgical interventions by setting standards for the conduct and reporting of both early and later phase trials .

**Contributors** BGM, NSB and JMB conceived the idea for the TORS review. BGM prepared the initial protocol draft with input from AG and PR and contributed to the revision of the manuscript. All authors contributed to the development of the idea and drafting and revision of the manuscript. JMB is the lead of the NIHR Bristol Biomedical Research Centre Surgical Innovation theme and formed the methodological ideas to understand surgical innovation with contributions from NSB, RH, BGM, SC, DE, BB, KNLA, NW and NH. All authors gave approval for the manuscript to be submitted.

**Funding** This study was supported by the NIHR Biomedical Research Centre at the University Hospitals Bristol NHS Foundation Trust and the University of Bristol. This work was undertaken with the support of the MRC ConDuCT-II (Collaboration and innovation for Difficult and Complex randomised controlled Trials In Invasive procedures) Hub for Trials Methodology Research (MR/K025643/1). JMB is an NIHR Senior Investigator. BGM and NSB are NIHR Academic Clinical Lecturers, and BB is an NIHR Academic Clinical Fellow.

**Competing interests** None declared.

**Patient consent** Not required.

**Provenance and peer review** Not commissioned; externally peer reviewed.

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
