## [Reviewer comments · BMJ Open]

ARTICLE DETAILS

TITLE (PROVISIONAL)	A protocol for the systematic review of the reporting of transoral robotic surgery
AUTHORS	Main, Barry; Blencowe, Natalie; Howes, Noah; Cousins, Sian; Avery, Kerry; Gormley, Alexander; Radford, Phil; Elliott, Daisy; Byrne, Benjamin; Wilson, Nicholas; Hinchliffe, Robert; Blazeby, Jane

VERSION 1 – REVIEW

REVIEWER	Meccariello Giuseppe Italy Morgagni Pierantoni Hospital University of Ferrara
REVIEW RETURNED	10-Oct-2017

GENERAL COMMENTS	Well Done.
------------

REVIEWER	Jason Chan Chinese University of Hong Kong Hong kong
REVIEW RETURNED	02-Dec-2017

GENERAL COMMENTS	This is well written and extensive protocol evaluating the reporting of TORS procedures in regards to its adherence to the IDEAL framework: Below are my comments regarding the protocol 1) If feasible it would be useful to be able to review the inclusion and exclusion criteria for this protocol that is mentioned that it will be collected in a customized form. Since this information will also be key in reviewing the protocol2) Is the data extraction form available for review as part of the protocol? This would be again useful to review despite the possible data to be extracted is mentioned. As this will be an essential part of the protocol3) To widen the catchment of papers would other regularly used search engines like pubmed, web of science and scopus be included?4) Given the protocol states that this is only examining TORS for the Oropharynx with oropharyngeal carcinoma and OSA the exclusion criteria should be expanded to exclude oral cavity, hypopharynx, parapharyngeal space that have also been described in the TORS literature.
--

	5) Given the narrow selection of TORS for OSA and Oropharyngeal carcinoma that is stated in the overall objective would benign lesions such as cysts/papillomas be excluded? This needs to be further clarified if there is a broader objective to include all benign oropharyngeal lesions.
--	--

VERSION 1 – AUTHOR RESPONSE

2. Reviewer 1 Comments – no comments to address

3. Reviewer 2 Comments

- i) Points 1 and 2 can be addressed by the publication of the data extraction form as a web appendix
- ii) These additional search databases have been added to the methods on page 7
- iii) Oral cavity, hypopharyngeal, and parapharyngeal cancers have now been added to the exclusion criteria on page 7
- iv) Cysts and papillomas have been added as example benign lesions on page 7.

I trust that these revisions meet your and the reviewers' expectations. Thank you for your further consideration of the manuscript .